# OpenReview forum: "Red Teaming for Trust: Evaluating Multicultural and Multilingual AI Systems in Asia-Pacific"
_ICLR.cc/2025/Workshop/BuildingTrust — BuildingTrust_

### Official Review · Reviewer_WrKJ · 2025-02-17
**Interesting work with substantial shortcomings**

**Rating:** 4
**Confidence:** 2

**Review:**

General:
- I think this paper uses an incorrect font type and size

Strengths:
- strong human-in-the-loop experiment with human subjects

Weaknesses:
- this paper reads like a report. While interesting insights, it misses an in-depth related work discussion and a stronger motivation.
- I am also missing a discussion on culture vs. language. Speaking a different language does not necessarily induce a different culture and vice versa.
- the proposed methodology is not very strong, nor novel. It is a very simple setup
- there is already quite some work on this, see XSafety and M-ALERT benchmarks, which have basically shown the same findings already.

---

### Official Review · Reviewer_MzEj · 2025-02-24
**Important contribution that would benefit from more procedural details and a clearer comparative conclusion**

**Rating:** 8
**Confidence:** 4

**Review:**

**Quality** (8): this study conducts a multi-lingual, multi-cultural red-teaming exercise to measure to what extent four LLMs exhibit biases in non-Western contexts. The motivation is well founded and this study addresses an important gap in AI safety with respect to both research and practice. While the writing and methodology is relatively clear, there is room for more explicit indications of processes and conclusions.

**Clarity** (8): The motivation and methodology for this study are relatively clear. The paper would be benefit from more clarity with respect to the human subjects and design, the annotation process, and the analysis/conclusions. In particular:

Human subjects and design:
-  In the paper body and/or in Appendix A1 - to understand implications and external validity of research, more details are needed about the demographics of participants, how they were recruited, how they were compensated, whether the points translated into an in-kind incentive, the time provided to them to play with models, their familiarity with models and previous use with LLMs, how many chances they were given to elicit biased responses, how many models each participant interacted with, whether they given access to outside tools while conducting this exercise?
- Authors might note in the paper body that instructions (A.2.7) give participants tips on how to elicit bias and as such the results in this paper aren’t representative of bias rates for average use (they may overestimate).
- Line 117: which models did each region receive? How were these chosen?
- Appendix A.2.3. what did participants actually see here? Why were model names broadly provided in the training but blinded in use? Why not blind in both?
- Lines 186 regarding “institutional guidelines,” did this study have an IRB? If so, could more details be provided?
- Authors might note in the paper body that instructions (A.2.7) give participants tips on how to elicit bias and as such the results in this paper aren’t representative of bias occurrence rates for average uses (they may overestimate).
- Line 136: what were differences in in-person and online formats? How were each conducted? When? Which types of participants were involved in each?

Annotation process:
- Line 139: “the platform automatically flagged potentially harmful prompts.” Did experts (line 123) only launch the two-stage review process for flagged prompts? If so, can the authors provide validation on the strength of the classifier?
- It seems important that the authors explicitly acknowledge that the “standardized rubric” is still quite subjective in that bias is still binarily determined by the annotator. The standardization comes in only with respect to how the prompt count maps to points; whether an exploit is unique; and how exploits aggregate across bias categories.
- Line 123: could the authors provide more information on what “expert” means in each context? How were the annotators recruited, what were their relevant demographics, and what constitutes cultural and linguistic proficiency?

Analysis/conclusions:
- 5.3 This section seems to diverge from the motivation of the paper. What is the hypothesis underlying this section? What larger point are these graphs aiming to make about Western v non-Western biases? Could this be made more explicit? Why were these particular examples chosen? Do similar trends exist for other biases and countries in the dataset? For Figure 1, if the implication is that these same results would not be observed in a western context, for example, could you show a Western reference point that contextualizes this gap?
- 5.2 how many submissions overall? Similarly, line 229 - why is percentage of *flagged* submissions rather than overall submissions?
- Table 2 “Key Focus Areas” — were these focus areas by design or is this what focus ended up being, without instruction?
- Table 4/5: it would be interesting to see data by model and country; by # of turns; and by paired results (the same prompt in English vs non-english). Does “success rate’ refers to biased responses out of flagged responses; or biased responses out of all responses? What does “Disp.” Mean?

Misc:
- Line 358 - could the authors clarify whether this is a limitation of the study methodology or of language models more generally? If the former, could the link be made more explicit?
- Line 973 - what does “domain experts” mean here?

**Originality** (8): this is a well-structured documentation of red-teaming in a novel context and adds original and scientific evidence to literature on bias of LLMs. Its scientific strength would be increased in addressing the above points on design, annotation, and analysis.

**Significance** (8): the paper shows us that bias exists in non-Western/English contexts for LLMs. However, to really drive home the significance of this research, the authors could consider adding clear reference points and cleaner comparative analysis. For example, is incidence of bias in these cases higher than in similar red-teaming exercises in Western contexts? When analyzing *pairs* of prompts, is bias higher in non-English versions than in English versions?

---

### Decision · Program_Chairs · 2025-03-04

**Decision:**

Accept

**Comment:**

The reviewers strongly disagree on the paper’s merits. One highly positive review (8) highlights the novelty, significance, and well-structured methodology, while another critical review (4, confidence 2) argues that the study lacks novelty and depth. Given the high confidence of the positive review and the low confidence of the negative one, the concerns raised should be weighed accordingly. The paper would benefit from a stronger related work discussion, clearer methodology framing, and deeper comparisons to existing benchmarks. Despite these limitations, this work addresses a complex and underexplored problem, making it a valuable addition to the workshop.